

# Factors influencing natural regeneration of *Fagus hayatae*

Chun Qin[1], Meng Tang[2] and Xue-mei Zhang[1]

[1] College of Life Science, Chian West Normal University, Nanchong, China
[2] Sichuan Xinhe Qingyuan Science and Technology Limited Company, Chengdu, China

## ABSTRACT

**Background**. Natural regeneration is pivotal for sustaining evolutionary processes in plant species. Identifying determinants that shape recruitment dynamics could elucidate key factors governing this critical biological process. However, the relationship between environmental variables and recruitment patterns in *Fagus hayatae* remains uninvestigated, despite its dual significance as a species endemic to China and a National Grade II Protected Plant. This knowledge gap persists even though understanding such ecological interactions could enhance conservation management for this vulnerable endemic tree species.

**Methods**. This study employed Pearson correlation analysis, redundancy analysis (RDA), and structural equation modeling (SEM) to examine environmental factors of natural regeneration in Fagus hayatae populations across ontogenetic stages (seedling, sapling, small tree) within three stratified elevational bands (low: 1,670–1,700 m; mid: 1,770–1,800 m; high: 1,890–1,940 m) of Micangshan Nature Reserve, located in Sichuan Province, China.

**Conclusions**. Structural equation modeling (SEM) revealed altitude-specific environmental regulation of *Fagus hayatae* regeneration dynamics. In low-altitude stands (1,670–1,700 m), the litter layer emerged as the primary limiting factor for seedling density (direct effect: −0.80), while diameter at breast height (DBH) suppressed sapling density through direct negative pathways (−0.63). Soil pH exhibited indirect positive mediation on seedling establishment *via* litter layer modification (0.42), suggesting synergistic management of litter removal and soil acidity regulation enhances regeneration success. Mid-elevation populations (1,770–1,800 m) demonstrated contrasting dynamics: DBH positively influenced sapling density (0.57), small tree density (0.89), and height growth (0.38), whereas available potassium indirectly reduced regeneration capacity through cascading effects involving canopy structure (DBH-mediated) and soil moisture, necessitating balanced canopy light penetration and weak acidic pH maintenance. High-altitude ecosystems (1,890–1,940 m) exhibited distinct nutrient-temperature synergies: available potassium exerted the strongest direct positive effect on small tree density (0.70), while air temperature (0.58) and litter thickness (0.53) jointly promoted vertical growth, highlighting the dual importance of organic layer preservation and thermal constraint mitigation in alpine forest regeneration. These findings delineate elevation-dependent multifactorial interaction networks governing *Fagus hayatae* population dynamics, establishing mechanistic frameworks for natural regeneration prediction while informing altitude-specific silvicultural interventions to optimize conservation outcomes.

Corresponding author
Xue-mei Zhang, 30049812@qq.com

# INTRODUCTION

Natural regeneration serves as a pivotal mechanism for generational succession in forest ecosystems (*Christopher, Deborah & Damien, 2005*; *Dech, Robinson & Nosko, 2008*; *Beckage, Lavine & Clark, 2005*; *Tinya et al., 2009*). This ecological process plays a crucial role in maintaining long-term forest sustainability (*Lewis et al., 2019*) and facilitates post-disturbance ecosystem recovery (*Pham et al., 2022*; *Shive et al., 2018*; *Johnson et al., 2021*). Compared to artificial regeneration, seed-based natural regeneration offers a more economical and ecologically efficient method for stand restoration (*Zhao, Sun & Gao, 2023*). The success of natural regeneration depends on complex interactions among aboveground factors (such as canopy structure, light availability, and litter dynamics), belowground factors (including soil nutrient stoichiometry and pH gradients), and climatic and topographic drivers (*Coomes & Grubb, 2000*; *Redmond & Kelsey, 2018*; *Liu et al., 2020*). These multidimensional regulators collectively determine regeneration success rates, forest resilience, and ecosystem functionality through direct physiological effects and indirect ecological feedbacks (*Coomes & Grubb, 2000*; *Johnson et al., 2021*; *Noguchi & Yoshida, 2004*; *Norghauer & Newbery, 2014*). Canopy density exerts differential effects on natural regeneration dynamics: studies have demonstrated positive correlations between canopy density and regeneration success (*Ádám, Ódor & Bölöni, 2013*; *Ali et al., 2019*; *Xu et al., 2017*), while other research confirms that severe light limitation suppresses pioneer species establishment (*Tinya et al., 2009*). Natural regeneration is also regulated by soil nutrient availability and physicochemical properties (*Wang et al., 2017*; *Bharathi & Prasad, 2017*). Under nutrient-poor conditions, soil moisture and nutrient availability may exert stronger control over seedling survival and growth than light regimes (*Coomes & Grubb, 2000*). In Guandi Mountain, the regeneration of *Larix principis-rupprechtii* exhibits particular sensitivity to soil pH (*Li et al., 2005a*). Furthermore, experimental evidence confirms that nitrogen (N), phosphorus (P), and potassium (K) act as limiting nutrients during stand regeneration processes, demonstrating dual regulatory effects on plant growth and development (*Mueller et al., 2013*; *Menge, Hedin & Pacala, 2012*). In addition to the aforementioned factors, litter thickness, air humidity, air temperature, soil temperature and a series of other factors exert significant influences on natural regeneration (*Liang & Wei, 2021*; *Hu et al., 2016*).

Elevation, as a comprehensive factor affecting natural regeneration, exerts strong control over seedling and sapling growth by regulating the spatial distribution of ecological factors including light availability, temperature regimes, water accessibility, and soil nutrient gradients (*Puhlick, Laughlin & Moore, 2012*; *Tinya et al., 2009*).

Environmental factors demonstrate phase-dependent dynamics throughout regeneration life cycles (*Wang et al., 2022*). *Pinus tabuliformis* regeneration exemplifies this pattern, requiring moderate shading during initial establishment phases but demanding

increased irradiance for subsequent developmental stages (*Wang et al., 2017*). Empirical evidence from *Pinus sylvestris* ecosystems documents differential regulation of seedling *versus* sapling densities by vegetation coverage, hydrological regimes, and organic horizon depth (*Mirschel, Zerbe & Jansen, 2011*). These findings underscore the necessity for phase-specific delineation of critical environmental thresholds to optimize target species regeneration.

*Fagus hayatae* Palib. ex Hayata (Fagaceae), a paleoendemic tree species restricted to China, is currently designated as a Class II National Key Protected Plant Species in China and classified as Vulnerable on the IUCN Red List (http://www.iucn.org/) (*Zhang, 2017*). Recent decades have witnessed progressive habitat insularization across its distribution range, a phenomenon attributed to synergistic effects of global climate warming, persistent anthropogenic disturbances, and intrinsic biological constraints including its naturally restricted biogeographic range and diminutive wild population sizes. This conservation crisis is further exacerbated by the species' observed low natural regeneration capacity (*Li et al., 2016a*; *Li et al., 2016b*), creating urgent challenges for population persistence. Notably, no comprehensive studies have systematically investigated the environmental determinants influencing natural regeneration processes in *F. hayatae* populations to date (*Li et al., 2016a*; *Li et al., 2016b*).

This study systematically investigated environmental drivers governing natural regeneration across distinct ontogenetic stages (seedlings, saplings, and small trees) in *F. hayatae* populations along three elevation ranges within the Micangshan Nature Reserve, Sichuan Province: low-elevation (1,670–1,700 m), mid-elevation (1,770–1,800 m), and high-elevation (1,890–1,940 m). The research objectives were threefold: (i) to evaluate critical environment factors influencing *F. hayatae* regeneration dynamics, (ii) to identify key environmental determinants regulating regeneration success across developmental stages, and (iii) to develop science-based management strategies for enhancing natural regeneration efficacy. These findings are expected to provide critical insights for optimizing *F. hayatae* population restoration and informing sustainable forest regeneration practices in temperate montane ecosystems.

## MATERIALS & METHODS

### Study site

The Sichuan Micangshan Nature Reserve (32°29′–32°41′N, 106°24′–106°39′E) occupies the northern margin of the Sichuan Basin. This protected area features a subtropical humid monsoon climate with elevational ranges spanning 1,500–2,000 m. Key climatic parameters include a 260-day frost-free period, mean annual temperature of 13 °C, annual sunshine duration of 1,355.3 h, and mean annual precipitation of 1,350 mm. The vegetation is dominated by subtropical evergreen-deciduous broadleaved mixed forests, with characteristic *F. hayatae* communities primarily occurring in deciduous broadleaved forests at 1,500–1,900 m elevation (*Li et al., 2016a*; *Li et al., 2016b*).

**Table 1 Sample plot information sheet.**

| Elevation classes | Sample plot name | Longitude | Latitude | Elevation | Slope | Slope direction | Canopy density |
|---|---|---|---|---|---|---|---|
| | Tianjiao | 106.553122 | 32.669247 | 1,679 | 49.8 | W-S | 62.9 |
| Low elevation | Wutan | 106.551783 | 32.672961 | 1,678 | 41.8 | W-N | 66.6 |
| | Luoshuidong | 106.551908 | 32.667725 | 1,681 | 35.72 | W-S | 67.1 |
| | Laolingou | 106.557317 | 32.658717 | 1,778 | 46.68 | N | 83.1 |
| Mid-elevation | Zhongshanbao | 106.557286 | 32.660628 | 1,781 | 57.28 | E-N | 67.4 |
| | Tabahe | 106.557933 | 32.656017 | 1785 | 47.2 | E-S | 66.9 |
| | Huangbailinya | 106.573794 | 32.657325 | 1,933 | 56 | N | 65.3 |
| High elevation | Huangbailin | 106.573131 | 32.658022 | 1,906 | 45 | W | 67.2 |
| | Miaojaiping | 106.572242 | 32.657089 | 1,922 | 45.35 | W | 67.8 |

Notes.
   W, west; W-N, northwest; W-S, southwest; E-S, southeast; E-N, northeast.

## Experimental design

In July 2022, three vertical transects were established along an elevational gradient within the Micangshan Nature Reserve in Sichuan. The vertical spacing between adjacent elevation bands was 100–130 m. The elevation bands were: low elevation band: 1,670–1,700 m; mid elevation band: 1,770–1,800 m; high elevation band: 1,890–1,940 m. The low-elevation site (1,670–1,700 m) was established to ensure coverage of the upper elevational limit of *Fagus hayatae* within the study area while comprehensively assessing regeneration constraints across the species' entire elevational gradient. Within each elevation band, three main plots (totaling nine plots; Table 1; Fig. 1), dominated by *F. hayatae*, were established, with a minimum distance of ≥1,000 m between adjacent main plots. Within each main plot, five 20 m × 20 m subplots were arranged in an X-shaped distribution pattern, with a spacing of >50 m between subplots. All *F. hayatae* individuals taller than two m were comprehensively surveyed within each plot; their diameter at breast height (DBH), tree height, and crown width were measured. Nested within each subplot, five 5 m × 5 m quadrats were established (maintaining the X-shaped distribution pattern) to monitor seedlings, saplings, small trees, and shrubs. Furthermore, nested within each of these 5 m × 5 m quadrats, five 1 m × 1 m herbaceous quadrats were placed, also maintaining the X-shaped distribution pattern.

To minimize precipitation effects on soil moisture, sampling was conducted ≥72 h post-rainfall. Within each nested quadrat, we removed the litter layer and collected 200 g soil samples (0–15 cm depth) following an X-shaped sampling pattern using a stainless-steel auger. We then homogenized soils from five quadrats to create composite samples were immediately stored in insulated containers and transported to the laboratory. Post-processing involved: Debris removal through manual sorting sieving (<1 mm mesh). Division into two aliquots: air-dried samples for physicochemical analyses (soil moisture, pH); Fresh samples preserved at 4 °C for enzymatic assays (*e.g.*, urease activity).

The environmental factors investigated in this study were categorized into four groups: geographic factors: slope (S) and elevation (A); stand factors: canopy density (CD), litter thickness (LT), shrub cover (SC), herbaceous cover (HC), number of adult trees (ANP), diameter at breast height (DBH); Climatic factors: light intensity (LI), air temperature

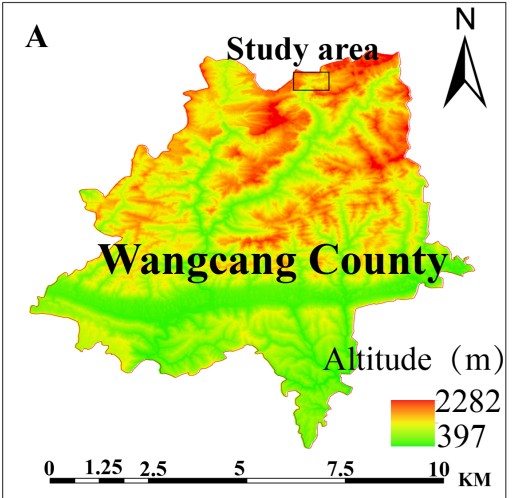
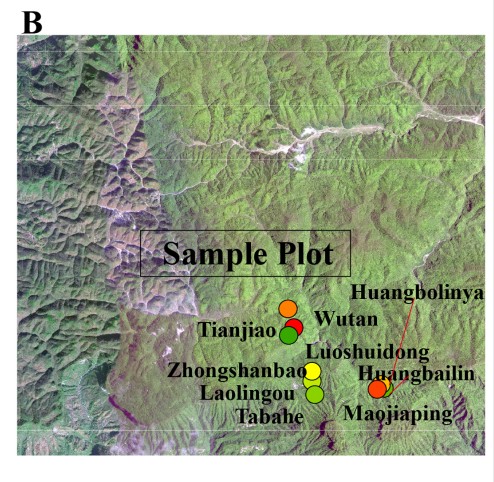

**Figure 1 Experimental plots.** The study site location in Wangcang County (A). (B) Sampling plots in Micangshan Nature Reserve.

(AT), soil temperature (ST), air humidity (AH), and soil moisture (SM); soil factors: pH, available phosphorus (AP), cation exchange capacity (CEC), total nitrogen (TN), soil organic content (SOM), soil water content (SWC), available potassium (AK), urease activity (UA), and catalase activity (CAT).

## Indicator measurement methods

Diameter at breast height (DBH) was measured at 1.20 m above ground level using a diameter tape. Geospatial parameters including elevation, longitude, latitude, and slope were recorded using a TX35-S300 GPS receiver (China) with satellite navigation timing (*Liang & Wei, 2021*). Litter layer thickness was measured following the vertical insertion method of a metal ruler to the soil surface (*Liang & Wei, 2020*). Canopy density was estimated through hemispherical photography analysis using the Canopy Capture application.

Vegetation coverage metrics were determined through systematic grid sampling: Shrub cover (SC) and herbaceous cover (HC) were quantified using a 10 × 10 grid system, calculated as the percentage of grid cells occupied by vegetation. Light intensity was measured using a Hipoint HR-350 sensor. Soil temperature (ST) and soil moisture (SM) were recorded between 10:00 and 16:00 local time using a TES-1365 thermohygrometer. Ambient air temperature (AT) and air humidity (AH) were measured with a Sunrise digital thermohygrometer.

Soil analyses followed standardized protocols: pH measurement using calibrated pH meter. Moisture content determination through constant-temperature drying (105 °C). Total nitrogen (TN) quantification *via* Kjeldahl digestion. Soil organic matter (SOM) analysis using dichromate oxidation. Cation exchange capacity (CEC) determination by sodium acetate method. Available phosphorus (AP) extraction *via* Olsen's sodium bicarbonate method (*Yan et al., 2015*). Available potassium (AK) measurement using
ammonium acetate extraction. Enzyme activities (urease and catalase) analyzed with microplate kits (Suzhou Keming Biotechnology).

## Data analysis

Since natural regeneration data (seedling density, SED; seedling height, SEH; sapling density, SAD; sapling height, SAH; small tree density, STD; and small tree height, STH) could not be standardized using conventional methods, nonparametric Kruskal–Wallis ANOVA was employed to analyze treatment effects, with raw data ranked to examine interactions among multiple factors (*Zhao, Sun & Gao, 2023*). These regeneration metrics were designated as response variables, while environmental factors were divided into four categories to represent the explanatory variable: geographic factors: slope (S), elevation (A); stand factors: canopy density (CD), litter thickness (LT), shrub cover (SC), herbaceous cover (HC), number of adult trees (ANP), and diameter at breast height (DBH); climatic factors: light intensity (LI), air temperature (AT), soil temperature (ST), air humidity (AH), and soil moisture (SM).

Soil factors: pH, available phosphorus (AP), cation exchange capacity (CEC), total nitrogen (TN), soil organic content (SOM), soil water content (SWC), available potassium (AK), urease activity (UA), and catalase activity (CAT). The relationships between environmental factors and *F. hayatae* regeneration were analyzed using CANOCO 5.0. Preliminary detrended correspondence analysis (DCA) of Fagus regeneration data revealed gradient lengths <3.0, indicating linear responses of this species to environmental variation, thereby justifying linear multivariate methods (*Li et al., 2018*; *Zhao, Sun & Gao, 2023*; *Liang & Wei, 2020*). Based on this, species-environment redundancy analysis (RDA) was implemented to identify dominant drivers impacting Fagus regeneration. The ordination focused on environment-regeneration correlations, with eigenvalue significance evaluated *via* Monte Carlo permutation tests (99% confidence) (*Chen & Cao, 2014*; *Chen, 2003*). Regeneration data were log-transformed to mitigate extreme value effects (*Gazer, 2011*), while environmental variables were square-root transformed prior to analysis to ensure variance homogeneity. Transformed datasets were analyzed using ANOVA, with Pearson correlation coefficients calculated to quantify associations between environmental factors and regeneration. Statistical analyses were performed in SPSS 22.0 (Chicago, IL, USA).

Structural equation modeling (SEM) is a robust multivariate technique for examining hypothesized causal relationships among latent variables through analysis of direct and indirect associations (*Zhao et al., 2019*). This method simulates multivariate interactions using two or more structural equations (*Gazer, 2011*) and visualizes complex variable relationships *via* intuitive network diagrams. SEM enables quantification of partial contributions from correlated variables, distinction between direct and indirect effects, identification of multiple influence pathways, and estimation of pathway strength comparisons (*Lv et al., 2023*; *Eisenhauer et al., 2015*).

In this study, we implemented SEM using Amos 26 (IBM SPSS, USA) to assess the effects of diverse environmental factors on six regeneration parameters: seedling density (SED), seedling height (SEH), sapling density (SAD), sapling height (SAH), small tree density (STD), and small tree height (STH). Path coefficients between factors were systematically

analyzed. The initial model underwent iterative refinement until achieving validation criteria: chi-square to degrees of freedom ratio ($\chi^2$/DF < 3), goodness-of-fit index (GFI > 0.90), comparative fit index (CFI > 0.90), root mean square error of approximation (RMSEA < 0.08), and non-significant $P$-values ($P > 0.05$) (*Zhu et al., 2022*; *Trivedi et al., 2016*). Final SEM diagrams were generated using Visio 2021 (Microsoft Corp., USA).

# RESULTS

## Distribution of *Fagus hayatae* at different elevations

We assessed the regeneration patterns of *F. hayatae* across ontogenetic stages (seedlings: mean height 0.2 m; saplings: 1.14 m; small trees: 2.96 m) along an elevation ranges. The census recorded 685 seedlings, 125 saplings, and 72 small trees, showing distinct elevational zonation.

Low-elevation zone (≤1,200 m): 132 seedlings, 34 saplings, 42 small trees. Mid-elevation zone (1,201–1,800 m): 480 seedlings, 62 saplings, 10 small trees. High-elevation zone (≥1,801 m): 73 seedlings, 29 saplings, 20 small trees.

## Regeneration correlation with environmental factors

The Pearson correlation analysis revealed significant associations between environmental factors and *F. hayatae* developmental stages (Fig. 2). Seedling density demonstrated strong negative correlations with litter thickness (−0.498, $P < 0.01$) and soil pH (−0.556, $P < 0.01$). Sapling density showed a significant negative correlation with soil moisture content (−0.376, $P < 0.05$). Small tree height exhibited positive correlations with available potassium (0.300, $P < 0.05$) and litter thickness (0.411, $P < 0.01$). Notably, soil water content was inversely associated with pH (r−0.369, $P < 0.01$).

## Environmental factors for RDA analysis and SEM

Redundancy analysis (RDA) was conducted to quantify the relative contributions of environmental factors to regeneration dynamics and evaluate variable associations within multidimensional data structures. This analytical approach enabled identification of optimal vegetation regeneration predictors through rigorous statistical frameworks. In the ordination diagram, the cosine values between environmental vectors indicate pairwise correlations, with this angular metric effectively projecting multivariate relationships onto a reduced-dimensional space. The geometric projection graphically represents the covariance structure between environmental variables and regeneration parameters. Cosine values approximating +1 reflect strong positive alignment with ordination axes, values approaching −1 indicate pronounced negative associations, and those near 0 demonstrate statistically insignificant relationships.

The RDA revealed that habitat factors collectively accounted for 68.1% of the variance across the first two ordination axes for *F. hayatae* density parameters (saplings, small trees, and seedlings; Fig. 3A). Regarding height parameters, the first two axes explained 47.2% of the cumulative variance (Fig. 3B). While inter-factor correlations showed concordance with Fig. 2 findings, the diffuse variable associations in RDA constrained precise identification of dominant drivers and their proportional impacts on density and height variations. To

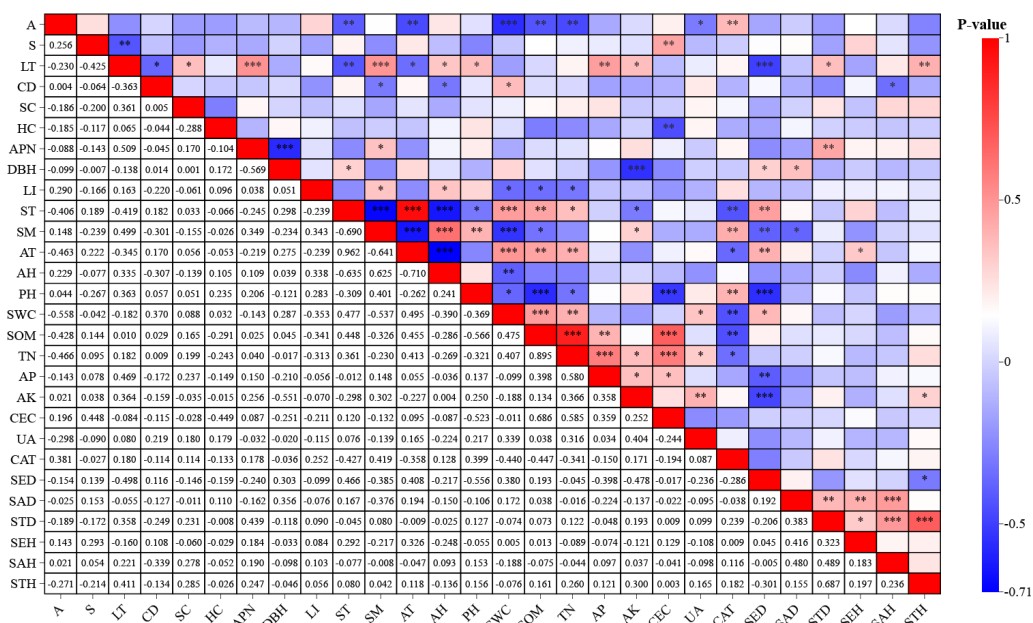

**Figure 2  Correlation analysis of environmental factors and regeneration.** A, altitudes; S, slope; CD, canopy density; LT, litter thickness; SC, shrub cover; HC, herbaceous cover; APN, number of adult plants; DBH, average diameter at breast height of mature plants; LI, light intensity; ST, soil temperature; SM, soil moisture; AT, air temperature; AH, air humidity; PH; SWC, soil moisture content; SOM, soil organic matter; TN, total nitrogen; AP, available phosphorus; AK, available potassium; CEC, cation exchange capacity; UA, urease activity; CAT, catalase activity; SED, seedling density; SAD, saplings density; STD, small tree density; SEH, seedling height; SAH, saplings height; STH, small tree height.

address this, we conducted Monte Carlo permutation tests to quantify environmental factor significance, with results integrated into Fig. 3's tabular summary. The contribution ranking of habitat factors to density variation emerged as: pH > cation exchange capacity > soil temperature. Key determinants of height development followed the hierarchy: litter depth > air humidity > soil water content.

Based on correlation analysis results, structural equation models (SEMs) were constructed by selecting environmental factors with high correlation coefficients and those demonstrating contribution rates exceeding 10% through Monte Carlo permutation tests. Environmental factors showing non-significant effects on regeneration were systematically eliminated according to established model fit criteria. Through iterative refinements *via* multiple modifications, seven environmental factors exhibiting the strongest associations with regeneration were ultimately retained. Subsequently, three stratified SEMs (low-, mid-, and high-elevation levels) were developed using elevation gradient as a covariate.

Low-elevation SEM (Fig. 4): litter thickness demonstrated the strongest negative direct effect on seedling density (−0.80), followed by available potassium's influence on sapling density (−0.46) and diameter at breast height (DBH) on small tree height (−0.63). While pH showed no direct effects on seedling density, it exhibited indirect positive effects through covarying factors (total effect: +0.32). These results establish litter thickness,

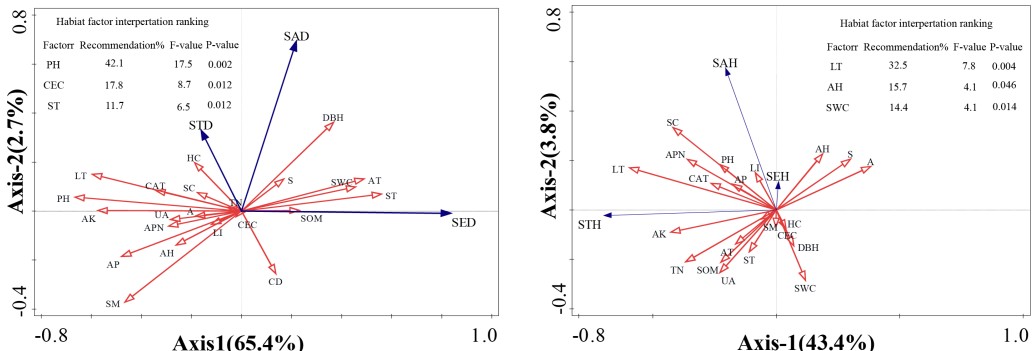

**Figure 3** **Ordination diagram of the redundancy analysis (RDA) results of environmental factors.** (A) The relationship between height and impact factors; (B) the relationship between density and impact factors. A, altitudes; S, slope; CD, canopy density; LT, litter thickness; SC, shrub cover; HC, herbaceous cover; APN, number of adult plants; DBH, average diameter at breast height of mature plants; LI, light intensity; ST, soil temperature; SM, soil moisture; AT, air temperature; AH, air humidity; PH; SWC, soil moisture content; SOM, soil organic matter; TN, total nitrogen; AP, available phosphorus; AK, available potassium; CEC, cation exchange capacity; UA, urease activity; CAT, catalase activity; SED, seedling density; SAD, saplings density; STD, small tree density; SHE, seedling height; SAH, saplings height; STH, small tree height.

available potassium, and DBH as pivotal drivers of *F. hayatae* regeneration in low-elevation habitats.

Mid-elevation SEM (Fig. 5): DBH displayed positive direct effects on sapling density (+0.46), small tree height (+0.57), and small tree density (+0.79). Contrastingly, negative direct effects emerged for pH on seedling density (−0.50), air temperature on small tree density (−0.41), and soil moisture on sapling density (−0.49). Available potassium manifested complex mediation pathways: while lacking direct effects on small tree density, it exerted substantial indirect negative influence through DBH mediation (total effect: −0.64), alongside indirect positive effects on seedling abundance (SAD) and small tree height (STH).

High-elevation SEM (Fig. 5): Available potassium emerged as the strongest positive predictor for sapling density (+0.70), followed by air temperature (+0.58) and litter thickness (+0.53) impacts on STH. A notable indirect relationship was observed where pH influenced small tree height exclusively through indirect pathways (total effect: +0.26), underscoring the complex interaction networks governing high-elevation regeneration dynamics.

## DISCUSSION

This study elucidates the complex interactions between environmental factors and natural regeneration dynamics across different developmental stages of *F. hayatae* along elevational gradients, revealing stage- and elevation-specific regulatory mechanisms. Our structural equation modeling (SEM) results highlight how key factors exert differential effects on regeneration outcomes at varying developmental stages and elevations, providing significant implications for conservation strategies.
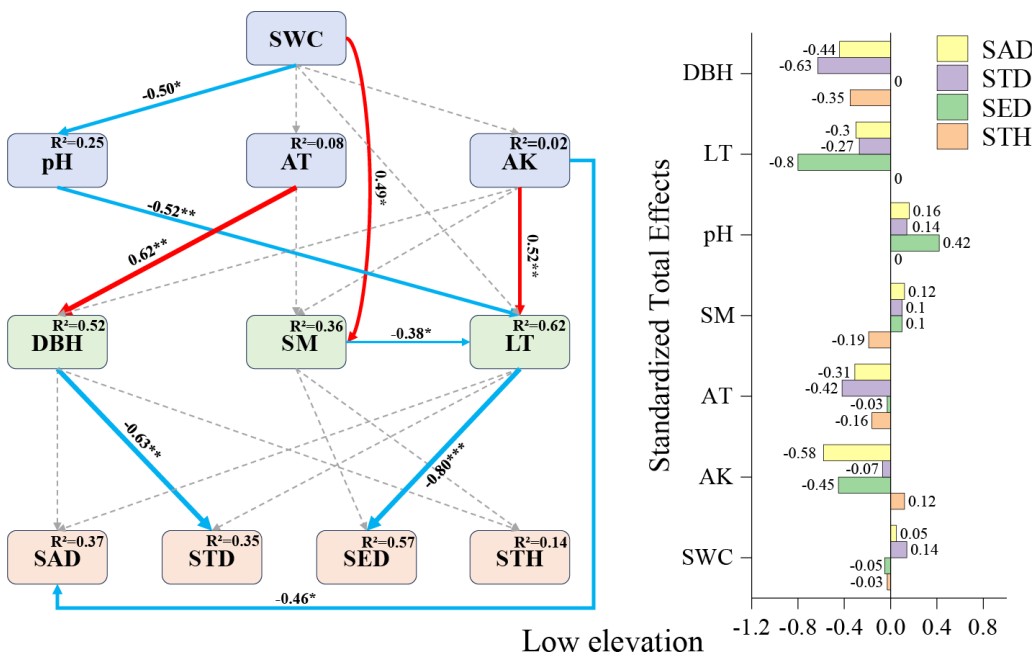

**Figure 4 Low-elevation Fagus hayatae updates standardized path diagrams of structural equation models.** Note: $\chi^2 = 32.260$; DF = 30; CFI = 0.964; RMSEA = 0.073; $P = 0.697$; $\chi^2$, chi-square value; DF, degree of freedom; RMSEA, root mean square error of approximation; CFI, comparison fit index; GFI, goodness of fit index; P, significance; the number on the arrow in the figure is the standardized path coefficient. The red arrow and the blue arrow represent the positive and negative effects, respectively. The solid arrow represents the significant path, the dotted arrow represents the insignificant path, and the arrow width and the path coefficient represent the strength of the influence. An asterisk (*) represents $P < 0.05$, two asterisks (**) represent $P < 0.01$, three asterisks (***) represent $P < 0.001$, SWC, soil water content; AT, air temperature; AK, available potassium; DBH, diameter at breast height; SM, soil moisture; LT, litter thickness; SED, seedling density; SAD, sapling density; STD, small tree density; STH, small tree height.

The transition from the seedling to sapling stage has been identified as a critical developmental phase in tree ontogeny (*Yan et al., 2015*). Our investigation quantified this population dynamic through demographic censuses of 685 *F. hayatae* seedlings, 125 saplings, and 72 small trees. The observed progressive reduction in abundance across successive life stages (seedling: sapling: small tree ratio = 685:125:72) provides quantitative evidence supporting this recognized developmental bottleneck.

### Factors affecting regeneration at low elevations

Excessively thick litter layers may inhibit seedling establishment through mechanisms such as light interception (*Wang & Zhang, 2008*), prevention of seed-soil contact (*Willis et al., 2021*; *Koorem, Price & Moora, 2011*), and release of autotoxic substances (*Willis et al., 2021*; *Pardos et al., 2007*). In this study, the litter layer exhibited the strongest direct negative effect on seedling density (−0.80) (Fig. 4), which aligns with previous findings on the mechanical barrier effect of litter accumulation (*Eckstein & Donath, 2005*; *Yu et al., 2017*). The negative effect of available potassium on sapling density (−0.46) (Fig. 4) may stem from a mismatch between soil nutrient supply and regenerating individuals'

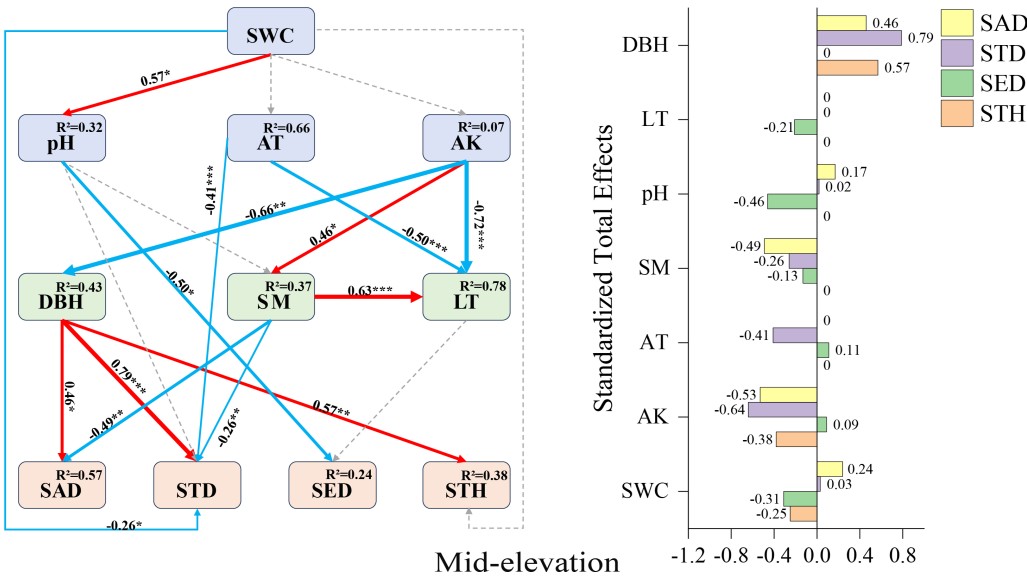

Mid-elevation

**Figure 5 Mid-elevation Fagus hayatae updates standardized path diagrams of structural equation models.** Note: $\chi^2 = 66.750$; DF = 32; CFI = 0.912; RMSEA = 0.069; $P = 0.069$; $\chi^2$, chi-square value, DF, degree of freedom; RMSEA, root mean square error of approximation; CFI, comparison fit index, GFI, goodness of fit index; P, significance; the number on the arrow in the figure is the standardized path coefficient. The red arrow and the blue arrow represent the positive and negative effects, respectively. The solid arrow represents the significant path, the dotted arrow represents the insignificant path, and the arrow width and the path coefficient represent the strength of the influence. An asterisk (*) represents $P < 0.05$, two asterisks (**) represent $P < 0.01$, three asterisks (***) represent $P < 0.001$, SWC, soil water content; AT, air temperature; AK, available potassium; DBH, diameter at breast height; SM, soil moisture; LT, litter thickness; SED, seedling density; SAD, sapling density; STD, small tree density; STH, small tree height.

demands. Specifically, when potassium released through litter decomposition fails to meet sapling growth requirements, soil available potassium shows a negative correlation with regeneration (*Li et al., 2021*; *Zhang et al., 2011*). In the low-elevation SEM (Fig. 4), although pH exhibited no direct effect on seedling density, it showed an indirect positive effect *via* the litter layer (0.42). *Shen & Zhao (2015)* demonstrated that soil microorganisms not only enhance nutrient uptake efficiency but also influence plants through nutrient mobilization and transfer. Therefore, we hypothesize that this phenomenon may relate to soil microbial activity, as elevated pH could stimulate proliferation of specific microbial taxa, accelerating litter decomposition. This process would reduce litter thickness and mitigate mechanical shading on seedlings (*Eckstein & Donath, 2005*; *Wang & Zhang, 2008*), thereby indirectly increasing seedling density. Similarly, the indirect negative effect of available potassium on seedling density *via* the litter layer (−0.45) (Fig. 4) may arise from antagonistic mechanisms: on the one hand, high available potassium may enhance litter input by promoting mature tree growth (*Zhang et al., 2011*), intensifying its physical obstruction on seedlings (*Koorem, Price & Moora, 2011*); on the other hand, potassium enrichment might inhibit microbial-driven litter decomposition (*Li et al., 2021*), exacerbating the litter layer's adverse effects. The SEM at low elevations revealed that air

temperature had no direct effect on sapling density but demonstrated an indirect negative effect *via* tree diameter at breast height (−0.42), highlighting a cascading interaction between climatic factors and canopy structure. A plausible pathway is that relatively higher air temperatures enhance mature tree growth, leading to increased trunk diameter and canopy closure (*Liu et al., 2020*), which subsequently reduces understory light availability (*Balandier et al., 2007*). Given that saplings require significantly more light resources than seedlings (*Wang et al., 2017*; *Zhao, Sun & Gao, 2023*), canopy shading may suppress sapling survival and establishment by limiting photosynthetic product accumulation, ultimately manifesting as an indirect negative effect of temperature elevation on sapling density. This finding parallels observations reported by (*Kovács, Tinya & Ódor, 2017*).

## Factors affecting regeneration at mid elevations

As shown in the mid-elevation SEM: the negative effects at mid-elevations primarily manifested as pH constraints on seedling density (−0.50) and soil moisture limitations on sapling density (−0.49). This aligns with (*Dyderski et al., 2018*), who reported reduced seedling survival under higher pH conditions, while we hypothesize that the inhibitory effect of soil moisture on saplings may relate to root hypoxia. Studies indicate that Fagaceae species require high light availability (*Alfaro Reyna, Martínez-Vilalta & Retana, 2019*). In our study, the positive effects of diameter at breast height (DBH) on sapling density (0.46), small tree density (0.79), and small tree height (0.57) likely originate from canopy openness regulating light conditions (*Liu et al., 2020*). Available potassium showed no direct effect on sapling density but exerted an indirect negative effect (−0.53) through two pathways: a negative effect *via* DBH (−0.66) and a positive effect *via* soil moisture (0.46). Similarly, available potassium had no direct effect on small tree density but generated an indirect negative effect (−0.64) through the same pathways: DBH (−0.66) and soil moisture (0.46). *Li et al. (2005)* demonstrated that mother trees with larger DBH typically acquire more soil water, light, and other resources to support seedling growth, thereby maintaining dominance in forests. As available potassium negatively affects mature trees, this indirectly reduces their support for saplings. Conversely, its positive effect on soil moisture may lead to hypoxia-induced root rot under elevated moisture levels. We hypothesize that available potassium suppresses the growth of saplings and small trees through these interactive pathways, ultimately decreasing their densities.

## Factors affecting regeneration at high elevations

The strong positive effect of available potassium on sapling density (0.70) at high elevations (Fig. 6) corroborates the classical theory of nitrogen, phosphorus, and potassium as limiting elements for forest regeneration (*Harpole et al., 2011*; *Wang et al., 2022*). The positive effect of the litter layer on small tree height (0.53) (Fig. 6) may stem from its water and nutrient retention functions (*Wang et al., 2022*). The facilitative effect of air temperature (0.58) (Fig. 6) underscores the critical role of thermal conditions in high-elevation regeneration (*Zadworny et al., 2021*). The indirect positive influence of pH on small tree height (0.26) (Fig. 6) further emphasizes the pivotal role of microbially mediated nutrient cycling in later growth stages (*Shen & Zhao, 2015*).
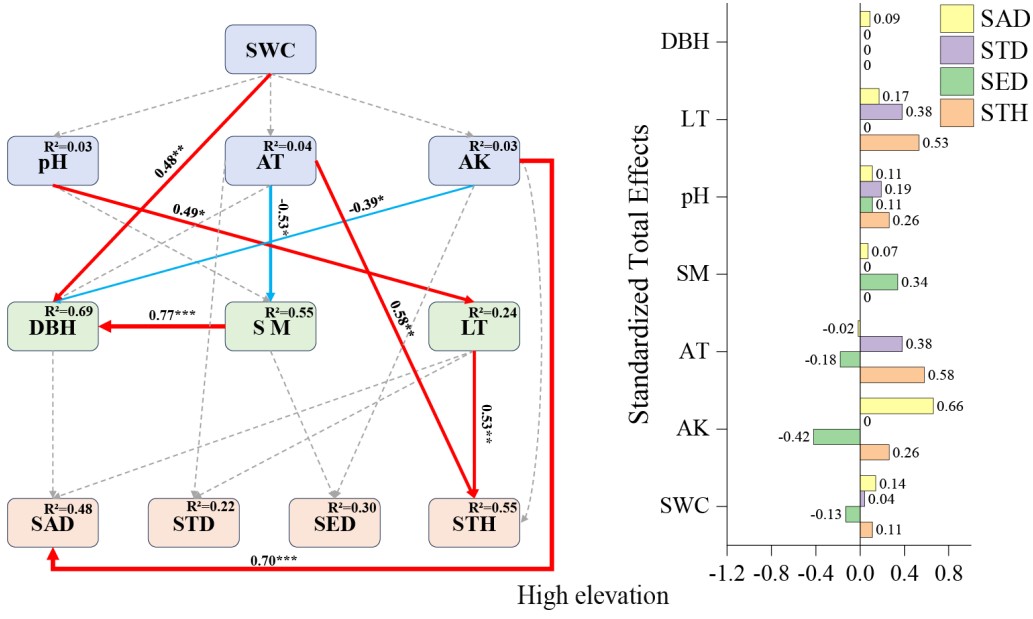

High elevation

**Figure 6 High-elevation Fagus hayatae updates standardized path diagrams of structural equation models.** Note: $\chi^2 = 29.870$; DF = 32; CFI = 1; RMSEA = 0; $P = 0.574$; $\chi^2$, chi-square value; DF, degree of freedom; RMSEA, root mean square error of approximation; CFI, comparison fit index; GFI, goodness of fit index; P, significance; the number on the arrow in the figure is the standardized path coefficient. The red arrow and the blue arrow represent the positive and negative effects, respectively. The solid arrow represents the significant path, the dotted arrow represents the insignificant path, and the arrow width and the path coefficient represent the strength of the influence. An asterisk (*) represents $P < 0.05$, two asterisks (**) represent $P < 0.01$, three asterisks (***) represent $P < 0.001$, SWC, soil water content; AT, air temperature; AK, available potassium; DBH, diameter at breast height; SM, soil moisture; LT, litter thickness; SED, seedling density; SAD, sapling density; STD, small tree density; STH, small tree height.

## Conservation recommendations for *Fagus hayatae*

This study elucidates the complex interactions between altitudinal environmental factors/drivers and natural regeneration dynamics of *F. hayatae*, revealing stage-specific and elevation-specific regulatory mechanisms. Our structural equation modeling (SEM) results highlight how key factors exert differential effects on regeneration outcomes across developmental stages and altitudinal gradients, providing critical insights for conservation strategies. To address key limiting factors for *F. hayatae* regeneration across altitudinal ranges, differentiated management strategies are required: low elevation (1,670–1,700 m): manually thin excessively thick litter layers to alleviate light interception and physical barriers, coupled with soil pH adjustment to enhance microbial decomposition. This accelerates litter degradation and mitigates inhibitory effects of available potassium accumulation on saplings. Additionally, reduce canopy closure to minimize light competition pressure on saplings. Mid-elevation (1,770–1,800 m): regulate canopy light transmittance to balance soil moisture and prevent root hypoxia. Adjust soil pH to weakly acidic ranges to improve seedling survival rates, while suppressing cascading negative effects of available potassium on saplings and small trees mediated through mature tree competition and soil moisture imbalance. High elevation (1,890–1,940 m): retain natural

litter layers to strengthen water and nutrient retention capacities. Implement microclimate interventions to mitigate low-temperature limitations, thereby enhancing thermal resource utilization for small tree growth.

## CONCLUSIONS

Understanding the environmental factors affecting the density and height of seedlings, saplings, and small trees is essential for achieving the regeneration of *F. hayatae*. Based on SEM analysis, our results indicate that the natural regeneration of *F. hayatae* (manifested as the quantity and distribution of seedlings, saplings, and small trees) is significantly influenced by distinctly different key factors across various elevation ranges, with overall regeneration capacity being weak and distribution uneven. Specifically: in low-elevation zones, litter (inhibiting seedling density), diameter at breast height (DBH) (inhibiting small tree density), and available potassium (inhibiting sapling density) are the primary factors constraining regeneration. In mid-elevation zones, DBH and available potassium play dominant roles; DBH promotes sapling density, small tree density, and small tree height to varying degrees, while available potassium inhibits these three metrics to varying degrees. In high-elevation zones, available potassium, litter, and air temperature emerge as the most critical factors influencing regeneration. Our findings may contribute to better management for the regeneration and sustainability of *F. hayatae* populations within Micangshan Nature Reserve.

### Funding

This work was supported by grants from the China West Normal University Research and Innovation Team (KCXTD2022-4). The funders had no role in study design, data collection and analysis, decision to publish, or preparation of the manuscript.

### Grant Disclosures

The following grant information was disclosed by the authors:
China West Normal University Research and Innovation Team: KCXTD2022-4.

### Competing Interests

The authors declare there are no competing interests. Meng TANG is employed by Sichuan Xinhe Qingyuan Science and Technology Limited Company.

### Author Contributions

- Chun Qin conceived and designed the experiments, performed the experiments, analyzed the data, prepared figures and/or tables, authored or reviewed drafts of the article, and approved the final draft.
- Meng Tang performed the experiments, analyzed the data, authored or reviewed drafts of the article, and approved the final draft.
- Xue-mei Zhang conceived and designed the experiments, authored or reviewed drafts of the article, and approved the final draft.

## Data Availability

The raw measurements are available in the Supplementary File.

## Supplemental Information

Supplemental information for this article can be found online at http://dx.doi.org/10.7717/peerj.19761#supplemental-information.

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
