# Peer review of "Factors influencing natural regeneration of Fagus hayatae"

_PeerJ, doi:10.7717/peerj.19761_

## Round 0.1 · original submission · Major Revisions

The reviewers provided various comments that should be addressed in a revised manuscript, including an important point about the name of the tree.
If submitting a revised manuscript, it needs to be carefully edited by an English language expert; as pointed out by the reviewer, the current English needs many corrections for improved clarity.

I have the following additional comments:

Title – I suggest “Factors influencing natural regeneration…..”
L 18 and elsewhere – I suggest referring to “recruitment” or “seedling recruitment” or “regeneration” rather than “renewal” throughout the manuscript.
L 27 – 31 These are Results, not Methods
L 32 – This heading could be “Conclusions” rather than “Results”? It seems interesting and valuable to briefly explain the contradictory findings that seedlings were negatively correlated with litter but sapling density and height were positively correlated with litter. What may cause this reversal of correlation between seedlings and saplings?
L 74 “elevation gradients” change to “elevation ranges”
L 136 “detrending correspondence” should say “detrending correspondence analysis”; It is not clear to me why a gradient length less than 3.0 indicates linear response to environmental change. Please cite a reference to support this claim.
L 142 Please add details on how the Monte Carlo approach was implemented.
L 143 This wording about SEM is not clear or accurate. Please reword and cite a reference to support your claim.
L 143 This statement does not seem accurate; please cite a reference to support this claim about SEM. In my experience, the main use of SEM is when you have identified suspected/expected causal relationships among variables. You would them build the SEM based on the suspected / expected causal relationships that you identified, and you can then check if you model has a high fit, which would provide support for your suspected / expected causal relationships.
L 146 “fitness” change to “fit”
L 163-206 This section repeats everything from Figure 2 and is too tedious to read. Please condense this section by discussing general patterns an only pointing out specifics for the most noteworthy relationships or relationships that directly relate to specific hypotheses that you identify in your Introduction.
L 228 What are “environmental silvers”?
L 244 Altitude per se does not directly impact organisms. Instead, factors that co-vary with altitude, such as temperature, might influence organisms directly.
L 262-264 If pH is negatively correlated with seedlings then lower pH means lower success of seedlings. Wording here referring to acidic pH being more favorable is confusing. Can you reword in terms of conditions that are not favorable (since it was a negative correlation)?
L 262-264 If you think that low pH directly causes reduced seedling recruitment then it could be valuable to include a graph showing the shape of the relationship between seedling density (y-axis) and soil pH (x-axis). It is relatively easy for restoration workers to measure soil pH and your graph could help them identify optimal sizes (or sites to avoid) for seeding or seedling planting in restoration areas.
L 280 “mobilization” change to “mobilize”
L 304 Is it possible that thicker litter around small trees could have been due to leaf drop from those small trees? Thus, litter may have been thinner at the time of seedling recruitment, but the litter layer has built up as the plants grew into small trees and leaves were dropped to the ground?
L 318-336 Canopy cover and Light intensity (LI) do not appear in your SEM. If you feel that these are important causal factors, then I think these factors should be included in your SEM. You SEM should portray you’re your most important expected causal links among variables.

Figure 1 – scale bars should be added to maps. The meaning of the colors on map A should be explained.

Figure 4 – the meanings of abbreviations need to be given in the caption, or the Figure 4 caption could refer to abbreviations given in the Figure 3 caption.

Figure 4 – This seems to show that your model is not significant (P = 0.697). Please check this or explain in the caption.

Figure 4 – What do the colors of the boxes mean? The color abbreviations need to be explained in the caption.

·

Basic reporting

This manuscript addresses the influence of elevation and microclimatic parameters on the success of early plant regeneration stages of Fagus hayatae. The objectives are worthy and can be evaluated with the planned experimental design. As the authors point out, and I agree, this work provides useful information for better understanding abiotic factors influencing natural regeneration of this species and, eventually, suggests adequate management practices, although this latter part would demand subsequent trials. Unfortunately, the manuscript itself has lower quality than the data. Countless formal issues depreciate the valuable effort of authors in taking good data and planning a sound experimental design. Just for citing a few of them, the style of citations and references is inconsistent; abbreviations of variables should be fully listed in the text (not only in legend of Figure 2); the writing is inaccurate and imprecise throughout the manuscript and, finally, English grammar and style should be deeply revised by a professional English translator. It is really a pity that so much effort and potential are lost in a careless manuscript. The good thing is that, at least, these formal aspects are solvable, but with a lot of work from the co-authors. I specify below suggestions to improve some passages.

The data in the Excel looks fine. However, readers cannot tell which variables are listed, as there is no list of variables with their abbreviations in an additional Excel sheet.

Experimental design

It is important that you clarify the number of soil samples that have been taken and how you sample. This information is in lines 100-103, but it is not clear. This is important, for example, to evaluate the validity of the SEM model; it seems to have too many path coefficients relative to the number of observations. Authors have measured many variables, but they have not made a rigorous, systematic description of the techniques, methodologies, and protocols used. This prevents readers (and reviewers) from evaluating the relevance and adequacy of the methods and reproducing the study. A better, more accurate description of the methodology is mandatory.

I do not agree with the use of elevation as a continuous variable, and with testing its effect through linear correlation analysis. To me, this is a categorical factor with three levels: High, Low, and Medium. I would analyze its effect in this way, using the exact altitude of each plot as a covariate.

Validity of the findings

In the Results, you over-describe the correlations. There is no need to mention every correlation; rather, just focus on the 5-10 most important correlations, the ones that you will later need for the Discussion. On the contrary, from the SEM model, you only describe and discuss the direct relations, ignoring the sequential connection of variables, and providing useful information about the indirect effects of SWC on the rest of the variables.

In the Discussion, the authors do not provide explanations for some of their results. A brief explanation of the relationship between soil pH and soil moisture with regeneration density is needed; these relationships are not obvious to me. On the other hand, I did not understand lines 275 to 286 explaining the negative relationship between soil K and P and seedling density.

Additional comments

The first part of the Introduction is too simplistic and untidy. I suggest reordering it, focusing on above vs below-ground factors, climatic vs edaphic factors, or any other similar way to make it tidier. And I would not just provide one side of the story; I mean, for example, concerning canopy density, you say it is positively related to regeneration, but there are multiple and different outcomes when studying overstory canopy effects and regeneration, depending on climate, forest history, or plant ontogeny. This same comment goes for the rest of the factors, or at least the more debatable ones. This should make the Introduction more thorough and focused. One specific issue I also see in the Abstract and Introduction is to state that regeneration favors diversity. This is not necessarily like this; think, for example, of dominating or invasive species that, through successful regeneration, prevent diversity. In fact, in Europe, European beech is a dominant species with low understory species richness. For example, we found through SEM modeling that litter accumulation below the canopy of beech had a clear negative influence on plant richness, across small spatial scales, supporting the role of litter thickness in structuring understory vegetation that you have observed.

Additional comments specified by lines (L):
L 90 Confirm this species is deciduous; it can be an evergreen in a deciduous forest.
L 96-97 Revise the use of terms “plots”, “sub-plots”, and “sites” as now they appear to be mixed up.
L 211: Explain, please, why the cosine value is used for this analysis.
L 216 I do not see the difference between panel 3a and b. The % of variance of 65.41% reported in the text does not seem to match that in Figure 3a.
L 228 “silvers”?

Cite this review as

Reviewer 2 ·

Basic reporting

Language is good, results are relevant. However, the paper needs more structure and organization,

Experimental design

Five sample plots of 20
98 m×20 m were set up in each sample plot

9 plots to cover how big an area? At the methods, it is confusing to understand the 9 plots.

Lines 104 to 109. How was this measured?
Line 113 - repeated words

Validity of the findings

lines 166 to 207: not an effective way to explain things
line 250: Just moisture would be affecting?
245 - What is improved pH?

Reviewer 3 ·

Basic reporting

This is an interesting short paper on biotic and abiotic factors affecting the regeneration of Fagus pashanica in Sichuan, China. Showing the importance of leaf litter for different developmental stages of beeches is particularly informative.

I made many small suggestions and corrections in the annotated PDF attached to my review.
One issue that should be fixed is the name of the species studied here.

Please note that Fagus hayatae has been treated as one species with two subspecies, subsp. hayatae from Taiwan and subsp. pashanica from China mainland.
Shen CF 1992: A monograph of the genus Fagus Tourn. ex L. PhD, City University of New York, New York, USA. 390 pp.
Alternatively, these subspecies have been treated as two distinct species, Fagus hayatae Palib. ex Hayata and Fagus pashanica C.C.Yang.
The latter concept is adopted by POWO (2025).
POWO (2025). "Plants of the World Online. Facilitated by the Royal Botanic Gardens, Kew. Published on the Internet; https://powo.science.kew.org/
E.g. https://powo.science.kew.org/taxon/urn:lsid:ipni.org:names:358600-1
This concept is also adopted in the most recent classification of Fagus (Denk et al. 2024).
Denk T., Grimm G. W., Cardoni S., Csillery K., Kurz M., Schulze E.-D., Simeone M. C. & Worth J. R. P. 2024: A subgeneric classification of Fagus (Fagaceae) and revised taxonomy of western Eurasian beeches. Willdenowia 54: 151–181. https://doi.org/10.3372/wi.54.54301

Therefore, I strongly recommend using the name Fagus pashanica instead of Fagus hayatae.
Moreover, the holotype of Fagus hayatae is from Taiwan, and the current study is from mainland China, Sichuan.

Experimental design

The experimental design is straightforward.

Validity of the findings

The validity of the findings is straightforward and can be helpful for forest managers.

Annotated reviews are not available for download in order to protect the identity of reviewers who chose to remain anonymous.

---

## Round 0.2 · Minor Revisions

L 24 “factors of natural regeneration” to “factors influencing natural regeneration”

L 91 This information needs clarification. The IUCN redlist seems to consider Fagus hayatae as Vulnerable, not Endangered: https://www.iucnredlist.org/species/31246/9619499

L 96 “inherently low natural regeneration capacity” – can you cite a reference to support his claim? Also, the term "inherently" seems problematic because it implies that this species could never have high regeneration under any possible conditions.

L 123 You specified that Fagus grows at 1500 m (L 118), can you briefly explain why your low elevation site did not start closer to 1500m?

L 128 “All woody plants taller than 2 m were comprehensively surveyed” – This implies that you measured all species of woody plants but the excel sheet only seems to show Fagus and there is no discussion of other species in Results or Discussion. Please clarify. See also my similar comment on L188 below.

L 136 “auger Homogenized” -- missing words?

L 188 here the text says that the RDA ordination focused on interspecies correlations, but the RDA as shown in Figure 3 seems to focus on environmental correlations rather than interspecies correlations. Please clarify here or in Results / figure(s) – I did not find discussion of interspecies correlations or any discussion other species.

L 213 “survival patterns” – Do you mean establishment patterns? Assessing survival patterns requires monitoring marked plants to assess their mortality over time.

L 255 “using altitude gradient as a covariate” – Please clarify. Do you mean that there were three independent models and within each model (low, medium or high), elevation was included as a covariate? I suggest "elevation" rather than "altitude".

L 259 “exhibited mediated indirect” to “exhibited indirect”
L 272 “mediated pathways” -- do you mean “indirect pathways”? Are you using “mediated” and “indirect” as synonyms? Using two different terms can be confusing.
L 365 How could you alleviate low temperature constraints? Perhaps by promoting additional global warming through continued burning of fossil fuels? But that does not seem like a responsible recommendation.

Table 1 – I suggest “elevation” instead of “altitude”.

---

## Round 0.3 · accepted · Accept

Thanks for your careful revision of your manuscript. I support publication of your revised manuscript.